# Biochemical Screening of Phytochemicals and Identification of Scopoletin as a Potential Inhibitor of SARS-CoV-2 M^pro^, Revealing Its Biophysical Impact on Structural Stability

**DOI:** 10.3390/v17030402

**Published:** 2025-03-12

**Authors:** Sarika Bano, Jyotishna Singh, Zainy Zehra, Md Nayab Sulaimani, Taj Mohammad, Seemasundari Yumlembam, Md Imtaiyaz Hassan, Asimul Islam, Sanjay Kumar Dey

**Affiliations:** 1Laboratory for Proteins and Structural Biology, Dr. B.R. Ambedkar Center for Biomedical Research, University of Delhi, Delhi 110007, India; sbano@acbr.du.ac.in (S.B.); jyotishnasingh08@gmail.com (J.S.); 2Centre for Interdisciplinary Research in Basic Sciences, Jamia Millia Islamia, Jamia Nagar, New Delhi 110025, India; zainyzehrajmi@gmail.com (Z.Z.); md186547@st.jmi.ac.in (M.N.S.); taj144796@st.jmi.ac.in (T.M.); mihassan@jmi.ac.in (M.I.H.); aislam@jmi.ac.in (A.I.); 3Laboratory for Proteins, Dr. B.R. Ambedkar Center for Biomedical Research, University of Delhi, Delhi 110007, India; seemasundari@gmail.com

**Keywords:** main protease, fluorescence quenching, FRET-based M^pro^ assay, isothermal titration calorimetry, Scopoletin

## Abstract

The main protease (M^pro^ or 3CL^pro^ or nsp5) of SARS-CoV-2 is crucial to the life cycle and pathogenesis of SARS-CoV-2, making it an attractive drug target to develop antivirals. This study employed the virtual screening of a few phytochemicals, and the resultant best compound, Scopoletin, was further investigated by a FRET-based enzymatic assay, revealing an experimental IC_50_ of 15.75 µM. The impact of Scopoletin on M^pro^ was further investigated by biophysical and MD simulation studies. Fluorescence spectroscopy identified a strong binding constant of 3.17 × 10^4^ M⁻^1^ for Scopoletin binding to M^pro^, as demonstrated by its effective fluorescence quenching of M^pro^. Additionally, CD spectroscopy showed a significant reduction in the helical content of M^pro^ upon interaction with Scopoletin. The findings of thermodynamic measurements using isothermal titration calorimetry (ITC) supported the spectroscopic data, indicating a tight binding of Scopoletin to M^pro^ with a K_A_ of 2.36 × 10^3^ M^−1^. Similarly, interaction studies have also revealed that Scopoletin forms hydrogen bonds with the amino acids nearest to the active site, and this has been further supported by molecular dynamics simulation studies. These findings indicate that Scopoletin may be developed as a potential antiviral treatment for SARS-CoV-2 by targeting M^pro^.

## 1. Introduction

SARS-CoV-2, responsible for the 2019 pandemic, shares similarities with SARS-CoV and MERS-CoV [1,2,3]. After infecting host cells, SARS-CoV-2 RNA translates two large polyproteins, pp1a and pp1ab, by encoding two overlapping open-reading frames (ORF1a and ORF1b) [4,5,6]. The papain-like protease (PL^pro^) and main protease (M^pro^) digest these polyproteins, which results in the production of 16 nonstructural proteins [6,7]. These NSP4-NSP16 proteins, which are produced after cleavage by M^pro^, play an essential role in replicating and transcribing the viral genome [8,9].

M^pro^, also referred to as 3-chymotrypsin like protease (3CL^pro^), is a cysteine protease comprised of around 300 amino acids. It is organized into three domains. This protease has a non-classical Cys–His catalytic pair (Cys145 and His41) located in the space between two domains, domains I and II. M^pro^ cuts polyproteins at 11 locations with high substrate selectivity and exhibits self-proteolytic activity [10]. Cys145 functions as a nucleophile in the Cys–His catalytic dyad, while His41 serves as a generic acid base in the proteolytic process. M^pro^ is entrenched within polyproteins as the nsp5 domain; therefore, it needs to employ its autoproteolytic activity to extract itself from these polyproteins and release the mature protease [11,12]. The active site of M^pro^ is made up of four sites (S1′, S1, S2, and S4), which often hold four fragments (P1′, P1, P2, and P3) of peptidomimetic inhibitors [4]. Its critical structural and functional attributes, along with unique characteristics like the absence of similar or homologous proteins in the human host, not only make M^pro^ crucial for the life cycle of SARS-CoV-2, but it also serves as an attractive target for antiviral drug development.

Although vaccines like Pfizer, Moderna mRNA-1273, ChAsOx1 (Covishield), and BBV152 (Covaxin) have been pivotal in combating SARS-CoV-2, they have certain limitations [13,14,15,16]. Their efficiency might be lower in particular groups, such as older individuals, immunocompromised people, or those with certain health issues who may mount weaker immune responses, making them more susceptible to infection [17,18]. Vaccines require time to produce immunity, which indicates they may not provide instant protection for those recently exposed to the virus [18]. Furthermore, the arrival of new variants may diminish the vaccines’ efficacy, requiring regular updates [19,20]. Drugs, meanwhile, are also essential to manage SARS-CoV-2 infection, especially for those individuals who have already been infected or might be at a higher risk of severe disease [20]. Unlike vaccines, drugs can be administered post-infection to reduce viral replication and symptoms, providing more direct protection to those already dealing with the disease [20]. Drugs are also flexible, i.e., they can be developed to target different stages of viral replication, which makes them more efficient in dealing with a wider range of virus variants [21]. For instance, in this study, we have targeted M^pro^, which is crucial for the poly-processing of polyproteins; thus, inhibiting it would help stop poly-processing and, alternatively, hinder viral replication [11]. Thus, it can be said that vaccines and drugs are both complementary in controlling the spread and impact of SARS-CoV-2 and have contrasting roles, being preventive and serving as a therapeutic option, respectively [16]. Therefore, developing drugs is crucial for people who cannot be vaccinated or who experience breakthrough infections and develop severe symptoms. Consequently, drugs provide critical support when vaccine limitations are encountered [16].

So far, some drugs like Veklury (remdesivir) and Lagevrio (molnupiravir) have been authorized or approved by the FDA but only under Emergency Use Authorization (EUA) against COVID-19 [22,23]. These drugs inhibit the RNA-Dependent RNA Polymerase (RdRp) protein [22,23]. However, M^pro^ offers various advantages over RdRp as an antiviral drug target. M^pro^ has a highly conserved sequence with low chances of mutation [24]. Although RdRp is also conserved, it shows slight variations in different coronaviruses [25]. This will ensure less chance of drug resistance developing through mutations in M^pro^ [11]. RdRp, on the other hand, may experience greater mutations due to direct involvement in viral RNA synthesis [26]. Using M^pro^ as a drug target provides another advantage, which is that it cuts polypeptides only after a glutamine (Gln) residue, thus unlike any human protease, M^pro^ has a unique cleavage selectivity, resulting in very low toxicity or side effects to host cells for M^pro^ inhibitors [11]. Therefore, even though there are effective RdRp inhibitors like remdesivir, the development of novel drugs with better efficacy and reduced side effects is still a challenge. Thus, M^pro^ has emerged as one of the most promising therapeutic targets for antiviral drug development [11,12]. Given the essential role of M^pro^ in the life cycle of the SARS-CoV-2 virus, targeting this protease with specific inhibitors presents a promising therapeutic approach [27]. Various M^pro^ inhibitors like N3, Ebselen, GC-376, boceprevir, lufotrelvir, nirmatrelvir, and coronastat have been identified and are currently under investigation for their ability to inhibit the replication of SARS-CoV-2 by disrupting the protease’s ability to cleave viral polyproteins [1,11,28]. However, none of these M^pro^ inhibitors have been approved by the FDA or any other drug controller across the world. Thus, there is still scope for identifying additional inhibitors of M^pro^.

Current research provides significant benefits in the search for antiviral treatments aimed at SARS-CoV-2, especially by utilizing phytochemicals. Due to their natural sources, most phytochemicals typically show lower toxicity compared to synthetic substances, making them safer treatment options with diminished chances of negative effects [29]. However, some phytochemicals have been reported to exhibit toxic effects, such as- strychnine, curare, and botulinum toxin, which can act on the nervous system [30]. Except for a few such examples, phytochemicals generally offer an intrinsic chemical variety, which is beneficial for structure-based drug discovery, enabling the identification of bioactive compounds that can selectively target viral proteins such as the M^pro^ of SARS-CoV-2, an enzyme crucial for viral replication. This research focuses on phytochemical inhibitors for M^pro^, targeting the existing shortage of drugs aimed at this crucial viral element, representing a significant progress in the therapeutic development of COVID-19. Moreover, this research improves pandemic readiness by establishing a foundation for creating natural, plant-derived substances that could be swiftly modified for upcoming viral dangers.

Gingerol, an essential component of ginger, has been acknowledged for its strong anti-inflammatory and antiviral effects, promoting immune health and exhibiting promise as a blocker of viral replication [31]. Pinene, a monoterpene found in pine trees and various plants, has demonstrated antimicrobial and antiviral properties that might improve host defense systems against viruses [32]. Limonene, a key constituent of citrus oils, is recognized for its antiviral, antioxidant, and immunomodulatory properties, potentially aiding in alleviating COVID-19 symptoms through the reduction of oxidative stress and inflammation [33]. A study by Neto et al. revealed limonene and pinene to be potential inhibitors or delta variants of SARS-CoV-2 [34]. Myrcene, typically present in plants such as hops and lemongrass, has potent anti-inflammatory and antioxidant effects, potentially offering a therapeutic advantage in addressing the inflammatory reaction to COVID-19 [35,36]. An in silico study by Ikanivic, Šeherčehajić et al., revealed that Scopoletin shows potential as an M^pro^ inhibitor for combating COVID-19, but this has not been verified by enzymatic assays or biophysical studies [37]. Another study conducted on *Artemisia annua* extracts by Baggieri et al., 2023, to investigate its virucidal, antiviral, and antioxidant properties against SARS-CoV-2 revealed that the fraction of extract containing Scopoletin (their study lacks an investigation of pure Scopoletin) may act as antiviral agent and provide support to combat variants of SARS-CoV-2 and other potentially evolving coronaviruses, thereby reinforcing its antiviral properties [38]. Thus, we directed our efforts toward phytochemicals, namely, Gingerol, Pinine, Limonene, Myrcene, and Scopoletin, since they have numerous advantages over synthetic small molecules. For example, plants play an essential role as a source of many therapeutics that play a vital role in human health, and these therapeutic properties are often attributed to phytochemicals [29]. Due to their natural origin, phytochemicals usually exhibit a lower toxicity, affect various viral and host pathways, minimize the chance of developing resistance, and have inherent anti-inflammatory and immunomodulatory characteristics, making them potential candidates for inhibiting SARS-CoV-2 [39,40]. We have employed biochemical and biophysical tools, like a FRET-based enzymatic assay, fluorescence spectroscopy, isothermal titration calorimetry, and CD spectroscopy, to verify the inhibition of M^pro^ by the best phytochemical/s screened in this study.

## 2. Materials and Methods

### 2.1. Virtual Screening of Selected Phytochemicals

Virtual screening was employed to predict and analyze the interactions between M^pro^ and five selected phytochemicals, namely Scopoletin, Gingerol, Pinine, Limonene, and Myrcene, based on evidence from literature studies as mentioned in the introductory section. Quercetin, which is a known inhibitor of M^pro^, was used as a positive control [34,38,41,42,43,44,45,46].

In the current study, molecular docking was conducted on M^pro^ (PDB ID: 6LU7) using five different compounds and one control, for the prediction of their binding affinities and detailed interactions. InstaDock, a one-click molecular docking tool, was used to perform the docking, and it automates the entire molecular docking-based virtual screening. QuickVina-W was employed for assessing the binding affinities of the ligand and protein [47]. Additionally, we have docked the best compound Scopoletin to SARS-CoV-1 M^pro^ (PDB ID: 7ZQW) and MERS-CoV (PDB ID: 9BOO) for a comparison.

The p*Ki* value, derived from the ∆G parameter, is calculated as the negative decimal logarithm of the inhibition constant using the following formula:∆*G* = RT (ln *Ki_pred_*)*Ki_pred_* = e ^(∆G/RT)^p*Ki* = −log (*Ki_pred_*)

Here, ∆*G* represents the binding affinity (J), with R (the gas constant) equal to 8.314 J mol^−1^ K^−1^; T (room temperature) is at 298.15 K, and *Ki*_pred_ represents the predicted inhibitory constant [47].

Ligand efficiency (LE) is often employed for identifying favorable ligands on the basis of a comparison of the average binding energy per atom values. The subsequent formula was used for computing LE:LE = −∆*G*/N
where LE stands for ligand efficiency (kcal mol^−1^ non-H atom^−1^), ∆*G* represents the binding affinity (kcal mol^−1^), and N denotes the number of non-hydrogen atoms present in the ligand [47].

#### Analysis of Molecular Interactions

The Discovery Studio Visualizer tool (https://discover.3ds.com/discovery-studio-visualizer-download (accessed on 11 November 2024) was utilized for examining the interactions taking place between the ligands and the protein.

### 2.2. Protein Expression

The SARS-CoV-2 M^pro^ sequence, containing a His-tag, was obtained from Genescript (Piscataway, NJ, USA) in a pET-28a (+) vector. Subsequently, this plasmid was transformed into the BL21 (DE3) Gold *E. coli* strain. A primary culture was started in 10 mL broth and kept for overnight incubation at 37 °C. The following day, a secondary culture was completed with 1 L media containing kanamycin. This culture was again incubated until the optical density of the solution reached 0.8 at 600 nm. Following this, isopropyl 1-thio-β-D-galactopyranoside (IPTG) was used to induce the culture for protein expression. This induced culture was incubated for 22 h at 16 °C. Following induction with IPTG, the cells were pelleted down for 15 min at 4 °C at 6000 rpm (using a Thermo Scientific multifuge X3R centrifuge, Waltham, MA, USA).

### 2.3. Protein Purification

To lyse the cells, lysis buffer with a composition of 50 mM Tris, 200 mM NaCl, 5 mM Imidazole, 1% Triton X-100, and 5% glycerol was added to the pellet and kept on a shaker incubator for about 15 min. After that, cell rupture was accomplished by sonication (using LABSONIC^®^ P homogenizer, LabWrench, Midland, ON, Canada) by performing 30 s ON and 30 s OFF cycles for 15 min. Following sonication, the cells were centrifuged again for 20 min at 9000 rpm and 4 °C for the collection of supernatants.

Affinity chromatography was used to purify the protein using a Ni-NTA column. Buffer with different imidazole concentrations (10 mM, 25 mM, 50 mM, 100 mM, 150 mM, and 250 mM) was added to the column one by one and the elution fractions collected. To check the proper folding and concentration of protein, the absorbance of the collected protein was measured using UV spectroscopy at 280 nm. The imidazole was then removed from the collected protein using dialysis. By utilizing the SDS-PAGE technique, the purity and expression of the purified protein were confirmed.

### 2.4. Western Blot Analysis

After purification, Western blot was used for detecting His-tagged protein. The protein was first purified using a His-tag. Samples were prepared with SDS loading buffer and denatured at 95 °C. The denatured samples were subsequently loaded onto a 10% SDS-PAGE gel for electrophoresis. After electrophoresis, the proteins were moved from the gel onto a PVDF or nitrocellulose membrane. To inhibit non-specific binding, the membrane was kept for incubation in 5% non-fat milk in TBST (Tris-Buffered Saline with Tween-20) for 1 h. The mixture was subsequently incubated with an anti-His primary antibody (Genescript, Piscataway, NJ, USA, Cat. #A00186S) at a 1:1000 dilution for 2 h at room temperature or overnight at 4 °C. Following washing with TBST to eliminate unbound primary antibodies, the membrane was treated with an HRP-conjugated (Horseradish Peroxidase) secondary antibody (Thermo Fisher Scientific, Waltham, MA, USA, Cat. # 31430) at a 1:10,000 dilution for 2 h at room temperature. A concluding wash with TBST was conducted to eliminate unbound secondary antibodies. The His-tagged protein was subsequently identified using a chemiluminescent substrate, the ECL reagent (Bio RAD, Hercules, CA, USA, Cat. #170-5060) and observed with an imaging device.

### 2.5. FRET-Based Enzymatic Activity Assay

A FRET-based enzymatic activity assay was performed following a STAR protocol published by Ihssen, Faccio et al. In our assay, we have utilized Ac-Abu-Tle-Leu-Gln-MCA (Cat/Code No.: 3250-v; Peptide Institute, Inc., Osaka, Japan, https://www.peptide.co.jp/en/ (accessed on 12 July 2023)), a specific fluorogenic probe/substrate similar to Ac-Abu-Tle-Leu-Gln-MCA, as described by Rut, Groborz et al. [48,49]. This FRET-based fluorogenic substrate consists of two natural amino acids, L-glutamine and L-leucine, and two non-natural amino acids, L-tert-leucine and L-2-aminobutyric acid. A fluorophore, 4-Methylcoumarin-7-amide (MCA), is connected to the C terminal of glutamine via a peptide bond. When the fluorogenic moiety is released due to cleavage by M^pro^, it produces a fluorescent signal, which can be measured by a Tecan instrument [50]. Thus, this protocol was followed to investigate the percentage of the activity of M^pro^ that Scopoletin could inhibit in a concentration-dependent manner. Initially, we validated the assay using both Quercetin (Sigma Aldrich, Missouri, USA) and nirmatrelvir (NMTV, a gift from the MedChemExpress, cat no.: HY-138687, https://www.medchemexpress.com/nirmatrelvir.html (accessed on 12 July 2024)). Our calculated IC_50_ values for the Quercetin and NMTV well corroborated the various published data [44,45,46,51,52], including those published by us previously for NMTV [6].

A 96-well black multi-well plate was utilized, in which M^pro^ and substrate were made to interact with each other in the presence of different inhibitor concentrations. Measurements were taken at a 380 nm excitation and 455 nm emission wavelength at a 380–455 nm wavelength range using a Tecan fluorimeter. Blank, positive, and negative controls were also used, where assay buffer served as a blank, substrate added to assay buffer served as a negative control, and M^pro^ and substrate added to assay buffer served as a positive control.

For the assay, 5 µM, 10 µM, 12.5 µM, 15 µM, 17.5 µM, 25 µM, and 50 µM Scopoletin were added to the wells of a 96-well black plate containing substrate mixed with assay buffer. To these wells, M^pro^ was added, and fluorescence intensity readings were taken immediately using a Tecan instrument. The % of inhibition was calculated using the 100 − (A_T_/A_C_ × 100) formula, where A_T_ represents the activity of the enzyme in the presence of Scopoletin, and A_C_ represents the activity of the enzyme in the absence of the Scopoletin. A concentration vs. % inhibition graph was plotted using these values. The Scopoletin concentration value at a 50% inhibition, abbreviated as the IC_50_ value, was calculated using the graph.

### 2.6. Fluorescence Spectroscopy

Understanding the protein–ligand binding mechanism is crucial for discovering and developing new therapeutic medicines [53,54,55,56]. Fluorescence quenching is a phenomenon where the fluorescence intensity of a protein decreases upon increasing the ligand concentration, if the ligand binds to the fluorophore-containing sites of the target protein [57].

Fluorescence spectroscopy was performed using a Jasco FP-6200 spectrofluorometer instrument to analyze the quenching of protein with the selected ligands (i.e., Scopoletin and Quercetin). The protein (12 µM) was titrated with ligand values ranging from 0.5 to 11 µM. The excitation wavelength was set at 280 nm, with an excitation band width of 5 nm and emission band width of 10 nm, and an emission scan between 300 nm and 400 nm with a peak at ~336 nm. The scanning speed was 250 nm/min with high sensitivity. All measurements were triplicated.

Intrinsic fluorescence studies were performed to investigate the protein–ligand interaction. Protein intrinsic fluorescence is determined by the surroundings of amino acids like tryptophan, tyrosine, and phenylalanine. Alterations in intrinsic fluorescence signify changes in the nearby surroundings of amino acid residues. Hence, we employed fluorescence measurements to investigate the impact of the ligand/compound on the M^pro^ structure and the structural modifications indicated by the emission spectrum.

#### Calculation of Binding Parameters

Fluorescence quenching in protein–drug complexes was analyzed mathematically with the Stern–Volmer and double logarithmic equations to determine their quenching and binding properties, as previously documented [58,59].(1)$$\frac{F0}{F}=1+Ksv\left[C\right]$$(2)$$\frac{log\left(F0-F\right)}{F}=logK+nlog[C]$$
where *F*0 represents the intensity of protein, while *F* represents the intensity of protein in the presence of the Scopoletin compound. [*C*] represents the variable concentration of the compound, and *K_sv_* represents the derived Stern–Volmer constant [60].

In order to determine the binding constants and number of binding sites, a double logarithmic graph (Equation (2)) was utilized. The *K* value represents the binding constant of the protein–compound complex, *C* represents the drug concentration, and “*n*” signifies the number of binding sites.

### 2.7. Thermodynamic Investigation Using Isothermal Titration Calorimetry (ITC)

The isothermal titration calorimetry method provides us with the thermodynamic information related to a binding reaction. For this study, a VP-ITC MicroCalorimeter instrument by MicroCal was used at CSIR-IGIB. The ITC was performed using 10 μM SARS-CoV-2 M^pro^ protein, which was titrated by injecting appropriate volumes of 300 μM stock solution of Scopoletin. The cell temperature was set at 25 °C, with the total number of injections being 30. The reference power was set at 10 μcal/s. With the exception of the first injection, which had a 5 μL volume, all the other injections were set at a 10 μL volume each. The stirring speed for mixing was set at 307 rpm, and the timing between each injection was 200 s.

### 2.8. ADMET Analysis of Scopoletin 

The pharmacokinetic and toxicological properties of Scopoletin were evaluated through an ADMET analysis using computational tools like SwissADME (http://www.swissadme.ch/, accessed on 16 January 2025). It was assessed for its potential to cause cancer, if any, and its overall pharmacological profile. Key ADMET parameters like absorption (gastrointestinal permeability and solubility), distribution (blood–brain barrier permeability), metabolism (interaction with CYP2D6Inh/subs), excretion, and toxicity (AMES) were predicted.

### 2.9. MD Simulation

A 100 ns MD simulation was conducted using GROMACS 2024.3 software, employing the Chemistry at Harvard Macromolecular Mechanics (CHARMM) forcefield. For the Scopoletin, topologies and force field parameters were developed using CgenFF (https://cgenff.com/, accessed on 16 January 2025). The Scopoletin–M^pro^ system was simulated in a virtual cubic box of water with a dimension of 10 Å and solvated using the gmx solvate module in the TIP3P water model. The TIP3P model is relatively simple, consisting of three interaction sites, which makes it computationally efficient. TIP3P is very compatible with the CHARMM force field, making it a versatile choice for MD simulation. Energy minimization was achieved through the steepest descent method, with 1500 steps carried out over 100 picoseconds. During the equilibration period, the temperature was raised from 0 to 300 K for both systems. Under periodic boundary conditions, the equilibration period lasted 100 picoseconds, maintaining a constant volume and a stable pressure of 1 bar. Both systems then underwent a final MD run lasting 100 ns, and the resulting trajectories were evaluated using the in-built tools of GROMACS.

### 2.10. Evaluation of Cytotoxicity by Scopoletin Using MTT Assay

An MTT assay was conducted to investigate the cytotoxic effect of Scopoletin. HEK293 (human embryonic kidney) cells were plated in each well of a 96-well plate for this purpose, and they were allowed to adhere and proliferate for 24 h. After adding the compounds, the cells were cultivated at 37 °C for 24 h. MTT solution (at a final concentration of 0.3 mg/mL of MTT) was added to the wells, and the cultures were incubated for an additional 3 h. With a 96-well plate reader, the absorbance at 570 nm was measured in each well. To determine the percentage viability, the growth of the treated and untreated cells was compared.

## 3. Results

### 3.1. Selection of Best Phytochemical, Scopoletin, Based on Virtual Screening

Five compounds selected after the literature review and they were studied by virtual screening. Quercetin was used as a known inhibitor of M^pro^ as a control. These compounds were docked and their binding energies were obtained (Table 1). After Quercetin, the highest binding free energy was observed for Scopoletin. and it displayed the formation of a bond with the H164 amino acid, which is a crucial part of the active site of M^pro^ (Table 1, Figure 1) [61]. Thus, Scopoletin was selected for further experiments.

### 3.2. Interaction Analysis

The interaction of compounds with the functionally significant residues in the binding pocket of SARS-CoV-2 M^pro^ were analyzed after docking (Figure 1 and Figure 2A,B). Figure 1 illustrates the detailed interactions between the selected compound, Scopoletin, showing that it engages in forming hydrogen bonds with the amino acids close to the active site. Table 2 lists the different interactions that Scopoletin forms with the amino acid residues TYR54, HIS164, HIS41, GLY143, ARG188, ASP187, GLN189, GLU166, MET165, HIS163, CYS145, and MET49 of M^pro^. The interactions are similar to the positive control, Quercetin, as both form a conventional hydrogen bond with TYR54, van der Waals forces/interactions with HIS41, ARG188, GLY143, ASP187, GLN189, GLU166, and MET165, and P-alkyl/alkyl bonds with MET49 and CYS145 residues (Figure 1 and Figure 2A,B and Table 2). Our results also indicated that Scopoletin can bind to the M^pro^ of SARS-CoV-1 and MERS (Appendix A), as evident from its strong binding energy of −5.1 and −6.7 kcal/mol, respectively.

### 3.3. M^pro^ Could Be Expressed and Purified Successfully

For checking the expression and purity, as well as to confirm the molecular mass of SARS-CoV-2 M^pro^, a 10% SDS-PAGE was performed. A single thick band was observed in a 100 mM imidazole elution fraction, indicating that the protein was >98% pure (Figure 3A). After comparing the ladder, the molecular mass of the protein was estimated to be around 33 kDa, which confirms that the protein purified was SARS-CoV-2 M^pro^. The further specificity of the protein was checked using Western blot by using an anti-His antibody (Figure 3B). A clear band was observed on the blot at a similar MW of 33 kDa, indicating the protein purified to be M^pro^ with a >98% purity.

### 3.4. FRET-Based Enzymatic Activity Assay Deciphers the IC_50_ of Scopoletin as 15.75 uM

Using the obtained FRET-based fluorescence intensity readings, the % inhibition was calculated, and a concentration vs. % inhibition graph was plotted. Using this graph, the value of IC_50_ was estimated by identifying the concentration at which 50% of protein activity is inhibited by Scopoletin (Figure 4). The calculated IC_50_ value for Scopoletin was 15.75 µM. Initially, we validated our assay of M^pro^ in the presence of a known inhibitor, Quercetin, and determined its IC_50_ to be 49.6 µM (Appendix A), which aligns well with previously published data of an IC_50_ 42.81 µM or IC_50_ 166 µM of the same control molecule [51,52]. We also conducted an activity assay of M^pro^ in the presence of another known inhibitor, NMTV, and our data matched well with our recently published separate work [6].

### 3.5. Fluorescence Quenching of M^pro^ with Inhibitor Scopoletin Indicates High Affinity

Fluorescence quenching is the process where the intensity of fluorescence of a protein decreases upon the addition of a ligand. It is mainly used to understand binding interactions and the calculation of binding affinity [62]. Fluorescence spectroscopy was employed to study the fluorescence quenching of M^pro^ on the addition of Scopoletin and a positive control, Quercetin. It was observed that Scopoletin could successfully quench the M^pro^ protein (Figure 5A,B). Further, the number of binding sites and binding affinity were calculated from a modified Stern–Volmer plot (Figure 6) and double reciprocal plot (Figure 7) [63]. The binding affinity was found to be 4.5116 M^−1^, with the number of binding sites being one. The R^2^ value was also calculated from the modified Stern–Volmer plot, which was 0.9755. The K_D_ value was estimated to be 0.75 × 10^5^ M. The K_SV_ value was evaluated as 9.32 × 10^5^ M^−1^ from a modified Stern–Volmer plot. The binding constant was calculated as 3.17 × 10^4^ M^−1^ using the modified Stern–Volmer equation (double logarithmic equation). The binding affinity of Quercetin was 5.7279 M^−1^, with the number of binding sites being one. The R^2^ value was 0.9765, with a K_D_ value of 0.12 × 10^6^ M. The K_SV_ value was evaluated as 1.12 × 10^6^ M^−1^ from a modified Stern–Volmer plot. The binding constant was calculated as 5.34 × 10^5^ M^−1^ using the modified Stern–Volmer equation (double logarithmic equation).

### 3.6. CD Spectroscopy Confirmed Changes in Secondary Structure of M^pro^ upon Binding of Scopoletin

Circular dichroism (CD) spectroscopy was conducted on M^pro^ in the presence of the ligand Scopoletin to analyze its effect on the SARS-CoV-2 M^pro^. The data suggested that the ligand binds to the protein, inducing structural changes, particularly in the secondary structure. These changes occur in a ligand (Scopoletin) concentration-dependent manner, indicating a strong interaction between the ligand and the protein. It was observed that upon addition of the ligand (Scopoletin of different concentrations), the secondary structure of SARS-CoV-2 M^pro^ was altered; however, its shape remained the same, which suggests a specific and localized binding interaction (Figure 8).

### 3.7. Isothermal Titration Calorimetry (ITC)

Isothermal titration calorimetry (ITC) is widely used to determine binding affinity and target engagement during drug discovery. A proper experimental design in ITC provides precise and valuable insights into drug–target interactions. ITC was used to determine the target engagement of Scopoletin and to estimate the dissociation constant for its interaction with M^pro^. It revealed the interaction between M^pro^ and Scopoletin to be an exothermic reaction (Figure 9). The thermodynamic parameters that were obtained are as follows: K_A_ = 2.36 × 10^3^ M^−1^, K_D_ = 4.24 × 10^−4^ M, ΔH = −1.8802 × 10^4^ kJ/mol (−4.494 × 10^6^ cal/mol), and ΔS = −15.0 kJ/mol K (−1.50×104 J/mol K). The Gibbs free energy calculated using the following equation was −1.4334 × 10^4^ J/mol (−3.426 × 10^6^ cal/mol). ITC was also performed with the positive control Quercetin, the values for which were K_A_ = 7.82 × 10^4^ M^−1^, K_D_ = 1.27 × 10^−5^ M, ΔH = −615.4 kJ/mol (−1.47 × 10^5^ cal/mol), and ΔS = −1.97 kJ/mol K (−471 cal/mol/deg). The Gibbs free energy calculated using the following equation was −1971.5 J/mol. The data for Quercetin well corroborated the previously published data [46].Δ*G* = Δ*H* – *T*;Δ*S*(3)

### 3.8. ADMET Analysis

After the docking screening, an assessment (in silico) of the ADMET profile of the phytochemical Scopoletin and its potential carcinogenicity were also conducted, to ensure its safety and non-carcinogenic nature (Table 3). The Ames test is a widely used assay to assess whether a compound causes genetic mutations (mutagenicity). If a compound shows no mutagenicity in the Ames test, it means it does not cause DNA mutations in the tested bacterial strains, indicating it is not likely to be a mutagenic risk [64]. The ADMET analysis showed that Scopoletin has a high GI absorption, meaning it is well absorbed in the digestive tract. It is not an OCT2 (Organic Cation Transporter 2) substrate, so it is not actively transported by the kidneys. Additionally, it tested negative in the Ames test, indicating no mutagenic effects. It suggested that Scopoletin is water soluble. However, it does not have CYP2D6Inh/Subs, meaning the compound neither inhibits nor acts as a substrate for drug-metabolizing enzymes.

### 3.9. MD Simulations Supported the Spectral Data at Dynamic Level

An MD simulation for 100 ns was performed on SARS-CoV-2 M^pro^ and M^pro^–Scopoletin complexes. Various structural and systematic characteristics of both systems were examined to determine their dynamics and stability in the solvent environment throughout the simulation period.

The structure of a protein can be altered significantly when a small molecule binds to it. The root mean square deviation, abbreviated as RMSD, is a key feature that is used to investigate the compactness and structural deviation of a protein [65].

The time evolution of their RMSDs was also calculated for M^pro^ and the M^pro^–Scopoletin complex during the simulations, and the average values were as 0.188 nm and 0.217 nm, respectively. The RMSD graph indicates that M^pro^ in an unbound state and M^pro^ in complex with Scopoletin remain stable during the simulation, which can be seen in Figure 10A. However, a slight rise in random fluctuations, reaching up to 0.15 nm, is noticed in the complexed M^pro^ between 0 and 40 ns. However, 40 ns later, the graph displays consistent and balanced RMSD values during the entire simulation. The RMSD fluctuations of the M^pro^–Scopoletin complex decrease after 75 ns and remain stable during the entire trajectory in comparison to the free M^pro^.

The mean fluctuation was analyzed for each residue and represented as a root mean square fluctuation (RMSF) to study the flexibility of residues in M^pro^ both in an unbound state and when bound to Scopoletin. Several residual fluctuations in different regions of M^pro^ were observed in the RMSF plot. As the simulation progressed at the region ranging from the N-terminus to the C-terminus, it was discovered that these fluctuations were stable and reduced upon Scopoletin binding (Figure 10B).

The conformational stability of M^pro^ both pre and post Scopoletin binding were assessed for both systems, using the radius of gyration (Rg) (Figure 11A). The calculated 2.2 nm average Rg values for M^pro^ were estimated to be the same before and after Scopoletin binding. The Rg plot indicated that binding with Scopoletin did not significantly alter the packing of M^pro^. Due to the packing adjustment of M^pro^, there may have been an initial fluctuation up to 40,000 ps to 60,000 of MD trajectories; however, the Rg reached a state of stability and equilibrium during the entire simulation, indicating complex stability.

The M^pro^ and M^pro^–Scopoletin complexes were found to have average solvent-accessible surface area (SASA) values of 150.95 nm^2^ and 149.26 nm^2^, respectively. In the case of both systems, the SASA plot demonstrates an equilibration pattern that is similar. The compact packing of the M^pro^ upon Scopoletin binding may be the cause of a slight decrease in the average SASA (Figure 11B).

#### 3.9.1. Dynamics of Hydrogen Bonds

A simulation was conducted to investigate the stability of M^pro^ prior to and following Scopoletin binding. Specifically, the time evolution of hydrogen bonds (H-bonds) formed within 0.35 nm was investigated (Figure 12A). The average number of intramolecular H-bonds in M^pro^ was calculated as 211 prior to binding and 212 following binding (Figure 13).

The probability distribution function (PDF) of the H-bonds was also plotted for both of the systems (Figure 12B).

#### 3.9.2. Principal Component and Free Energy Landscape Analyses

Principal components analysis (PCA) is an effective technique for identifying the dominant modes of motion in protein. Determining the protein configurational space, which carries a few degrees of freedom, aids in the explanation of how the motion occurs.

Through the method of essential dynamics, we examined how the M^pro^ and M^pro^–Scopoletin complex sample different conformations using PCA. Figure 14A,B display the conformational sampling of the M^pro^ and M^pro^–Scopoletin complex in the necessary subspace. The projection of M^pro^ shows conformational sampling with the EV1 and EV2 projected by the Cα atoms of protein. The M^pro^–Scopoletin complex was found to share the same conformational subspace as the free M^pro^. Additionally, Figure 14C,D demonstrate the time evolution of projections of trajectories on both EVs and residue fluctuations of M^pro^ on EV1.

The FEL analysis offers an atomic-level insight into a protein–ligand bound system, possible metastable states, and binding transition states, which may be used to design an inhibitor. To investigate the native states and conformational stability of the M^pro^ and M^pro^–Scopoletin complex, the first two PCs were utilized to generate the FELs. Figure 15A,B show the contoured FELs of the M^pro^ and M^pro^–Scopoletin complex, respectively. A darker shade of blue is suggestive of conformational states that are closer to native states and have lower energy levels when examining the plots. We found that M^pro^ is limited to three local basins and has only one global minimum. Similarly, M^pro^ acquires distinct states with multiple minima when Scopoletin is present, displaying three local basins with various conformational motions (Figure 15B)

### 3.10. MTT Assay of Scopoletin to Evaluate Its Cytotoxicity in HEK293 Cells

An MTT assay was performed to investigate the cytotoxic effects of Scopoletin in HEK293 cells. Quercetin was used as a positive control. The findings indicated that Scopoletin maintained acceptable cell viability and were comparable to those of Quercetin. The results are available in the Appendix A.

## 4. Discussion

This study aimed to explore the inhibitory potential of Scopoletin, a bioactive phytochemical, on SARS-CoV-2 M^pro^ by utilizing biophysical, biochemical, and in silico methods. The results demonstrated a strong binding affinity of Scopoletin towards M^pro^, significantly altering its structure and inhibiting its enzymatic activity. The results indicated the potential of Scopoletin as a viable antiviral compound targeting M^pro^, which is a critical enzyme in the SARS-CoV-2 life cycle.

The molecular docking analysis displayed that Scopoletin interacts directly with the key residue within the active site of M^pro^, H164, that is located within the active site and participates in the catalytic activity of the protease, along with C145 and H41 [66,67]. On binding at this site, Scopoletin likely disrupt the ability of the M^pro^ enzyme to process viral polyproteins, eventually inhibiting viral replication. Further validation of this interaction was achieved by a FRET-based enzymatic activity assay, which demonstrated that Scopoletin inhibits M^pro^ and has an IC_50_ value of 15.75 µM. Fluorescence quenching studies further confirmed the stable complex formation between M^pro^ and Scopoletin, with a binding constant of 3.17 × 10^4^ M^−1^ and a 1:1 binding stoichiometry. These results are indicative of a strong and specific interaction of Scopoletin with M^pro^, thus resulting in the effective inhibition of the activity of M^pro^.

Structural aspects were studied using CD spectroscopy, which revealed the binding of Scopoletin leads to a significant disruption of the helical content of M^pro^ without completely denaturing the enzyme, indicating a partial structural modification that is sufficient to impair its function. This was supported by the results of the MD simulation, where the root mean square deviation (RMSD) analysis showed the stable formation of the M^pro^–Scopoletin complex throughout the 100 ns simulation, with an average RMSD of 0.217 nm compared to 0.188 nm for free M^pro^. During the first 40 ns, a slight increase in RMSD is seen, which indicate initial structural adjustments upon the binding of Scopoletin; however, the system stabilizes later, indicating that the complex is structurally stable over time.

A decrease in residual flexibility was observed in the root mean square fluctuation (RMSF) analysis across various regions of M^pro^ particularly near the N- and C-termini, on binding with Scopoletin. This reduced flexibility suggests that the binding of Scopoletin restricts the movement of key regions that are involved in the enzymatic activity of M^pro^, further contributing to its inhibitory effect.

Isothermal titration calorimetry (ITC) was utilized to gain thermodynamics insights by studying the binding interactions between M^pro^ and Scopoletin, revealed an exothermic binding reaction with a dissociation constant (K_D_) of 4.24 × 10^−4^ M, which suggests a relatively high binding affinity. The negative enthalpy of (ΔH = −1.8802 × 10^4^ kJ/mol) indicates that the interaction is stable and specific, primarily driven by hydrogen bonding and van der Waals interactions. The negative Gibbs free energy (ΔG = −1.4334 × 10^4^ J/mol) further confirms a spontaneous and energetically favorable binding process.

The radius of gyration (Rg) plot from the MD simulations showed the overall compactness of M^pro^ remains relatively unchanged upon the binding of Scopoletin, with an average Rg of 2.2 nm for both the free and complexed forms of M^pro^. This suggests that while Scopoletin induces local conformational changes, it does not significantly affect the overall packing of M^pro^. Similarly, the solvent-accessible surface area (SASA) values for M^pro^ and the M^pro^–Scopoletin complex were 150.95 nm^2^ and 149.26 nm^2^, respectively, indicating a slight reduction in solvent exposure due to tighter packing in the complexed state.

The MD simulations also monitored the number of hydrogen bonds within M^pro^. The time evolution of hydrogen bonds showed that the average number of H-bonds remained almost constant, with 211 H-bonds in the free state and 212 H-bonds in the M^pro^–Scopoletin complex. This stability indicates that Scopoletin binding does not significantly disrupt the intra-protein interactions with M^pro^. This analysis was further supported by the probability distribution function (PDF) analysis of H-bonds, showing a similar distribution pattern for both systems.

The collective motions of M^pro^ and the M^pro^–Scopoletin complex were examined using principal component analysis (PCA). The 2D projection of trajectories on the first two eigenvectors (EV1 and EV2) indicated that the complex samples similar conformational subspaces as free M^pro^. However, the presence of Scopoletin introduced additional conformational flexibility, as evidenced by the distinct conformational basins observed in the free energy landscape (FEL) analysis. While free M^pro^ exhibited only one global minimum, the M^pro^–Scopoletin complex occupied multiple local basins, indicating a more diverse set of low-energy conformations. This suggests that Scopoletin binding induces new conformational states in M^pro^, potentially associated with the inhibition of its proteolytic activity.

To evaluate the safety profile of Scopoletin, its cytotoxicity was analyzed using an MTT assay, as it is a critical parameter for drug development. The findings revealed that Scopoletin consistently maintained acceptable cell viability and remained stable even at higher concentrations, indicating minimal disruption to the cells and an endurable safety margin. Scopoletin displayed comparable cytotoxicity levels to that of the control, Quercetin, indicating that further development can be directed towards Scopoletin to optimize its therapeutic efficacy (Appendix A).

### Scopoletin, a Phytochemical, Is a Potential Inhibitor of M^pro^ and Suitable for SARS-CoV-2 Treatment

Scopoletin, identified in this research via biochemical analysis, demonstrates considerable potential as an inhibitor of the SARS-CoV-2 main protease (M^pro^), an enzyme essential for viral replication. Scopoletin, a natural coumarin, was recognized as an M^pro^ inhibitor via virtual screening and subsequent enzyme assays. It displayed an IC_50_ of 15.75 µM, signifying its effectiveness in inhibiting viral replication by directly attaching to the protease and impairing its function. Fluorescence spectroscopy investigations revealed that Scopoletin attaches with strong affinity to M^pro^, resulting in fluorescence quenching and alterations in the M^pro^ structure. Moreover, synchronous fluorescence and circular dichroism (CD) spectroscopy indicated changes in the helical configuration of M^pro^, implying a significant interaction and possible conformational destabilization of the viral protease. 

Scopoletin is a white crystalline phytochemical that dissolves in organic solvents such as methanol and chloroform [41]. Though the dosage details for COVID-19 remain uncertain, earlier research involving comparable coumarins indicates that micromolar range doses might be therapeutically beneficial [68]. Nonetheless, clinical trials are essential to verify safe dosing schedules in humans. Not much data are available on the adverse effects of Scopoletin in antiviral uses. Nonetheless, it has been utilized in conventional medicine with minimal side effects, mainly demonstrating safety at reduced doses. Additional toxicology research is crucial to determine a safety profile for the use of high-dose antivirals. Although Scopoletin is not approved for viral infections, traditional medicine has investigated it for its antiviral, immunomodulatory, anti-inflammatory, and anti-oxidant effects and many more such properties, which are listed in Table 4 [68,69]. The extract of *Artemisia annua* containing Scopoletin has also been studied for its antiviral effects [38]. Due to its encouraging ability to inhibit SARS-CoV-2 M^pro^, Scopoletin deserves additional clinical investigation as a possible candidate for treating COVID-19.

## 5. Conclusions

This study demonstrates that Scopoletin is a potent inhibitor of SARS-CoV-2 M^pro^, exhibiting strong binding affinity, inducing conformational changes, and effectively inhibiting the enzyme’s activity (Figure 16). It displays various biological activities like anti-inflammatory, antiviral, and anti-oxidation ones, and many more that are mentioned in Table 4. The combination of molecular docking, MD simulations, fluorescence quenching, CD spectroscopy, and ITC provides a comprehensive understanding of the inhibitory mechanism of Scopoletin. The MTT assay (Appendix A) provides the safety profile of Scopoletin, which suggests it to display acceptable cytotoxic levels, supporting its potential as a therapeutic candidate for further development. In most of above experiments, Quercetin was used as a control. Our findings are also supported by Yuan et al.’s study on Scopoletin for a specific cancer [69]. The reduced flexibility, stable binding, and thermodynamic properties of the M^pro^–Scopoletin complex, and its favorable safety profile, suggest that Scopoletin or its derivatives can serve as antiviral agents targeting M^pro^, offering a promising lead for further drug development. Future in vivo studies are necessary to validate these findings and explore the clinical efficacy of Scopoletin in treating COVID-19.

## Figures and Tables

**Figure 1 viruses-17-00402-f001:**
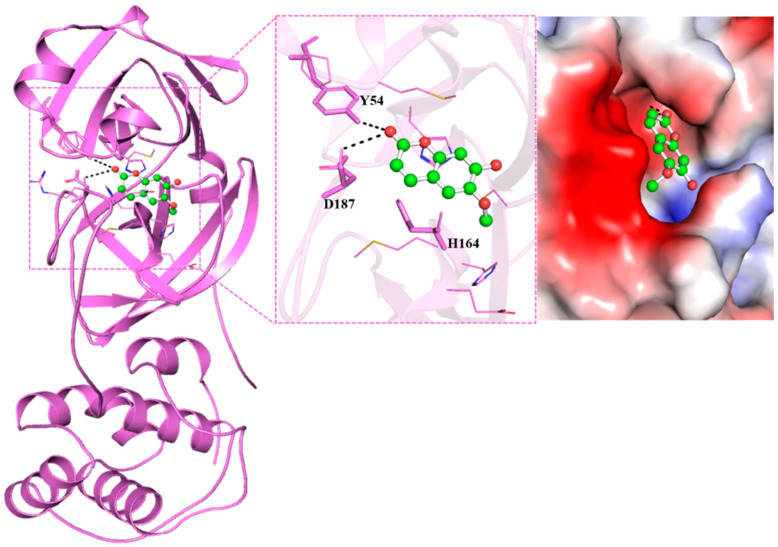
Structural depiction of docked compound, Scopoletin, within the binding cavity of SARS-CoV-2 M^pro^, depicting interactions with Y54, D187, and H164.

**Figure 2 viruses-17-00402-f002:**
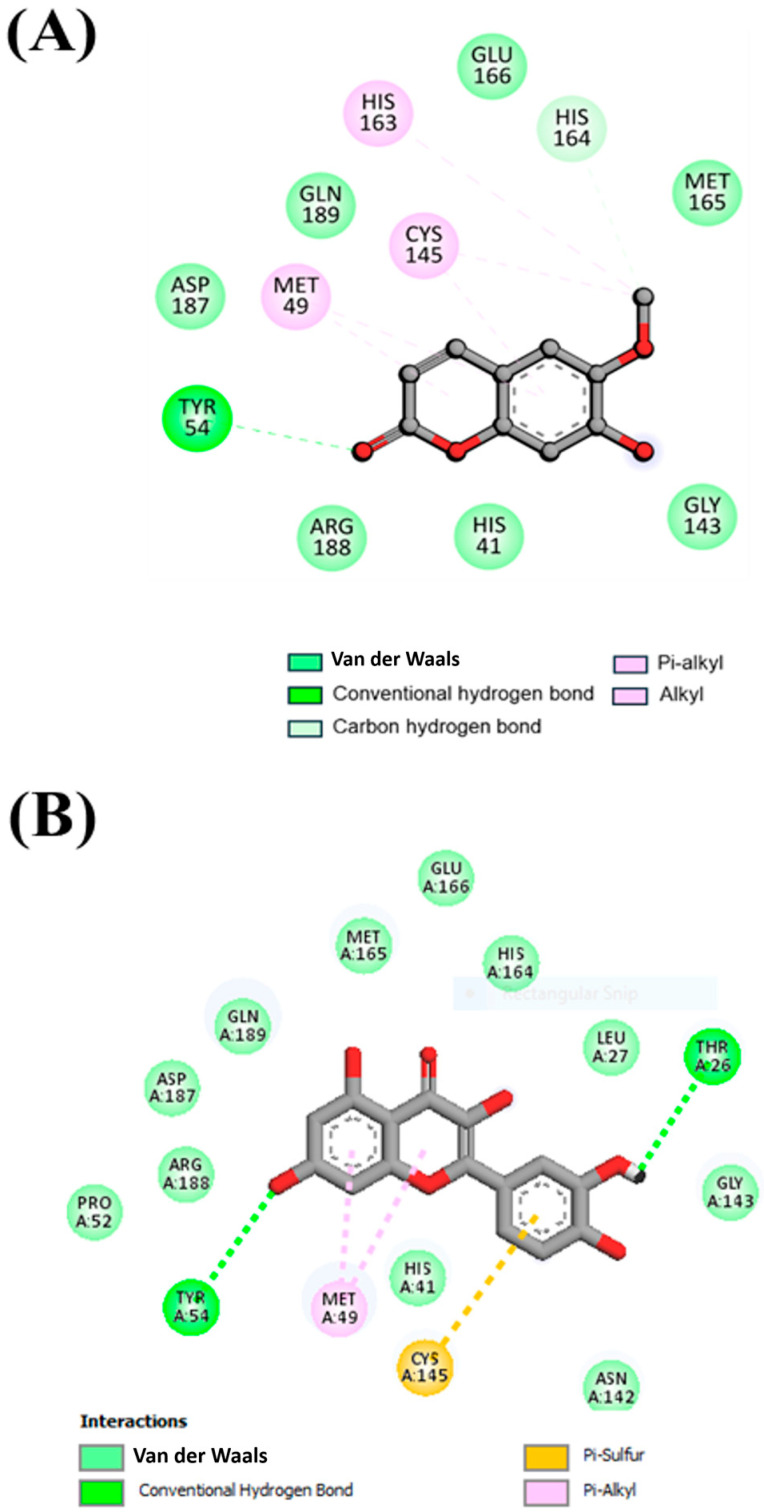
(**A**) A 2D graphical depiction of the interactions present between the selected compound, Scopoletin, and the residues of SARS-CoV-2 M^pro^. (**B**) A 2D graphical depiction of the interactions present between the positive control, Quercetin and the residues of SARS-CoV-2 M^pro^, depicting HIS41, GLY143, ARG188, ASP187, GLN189, MET165, CYS145, HIS164, and TYR54 residues as the key amino acids involved in these interactions.

**Figure 3 viruses-17-00402-f003:**
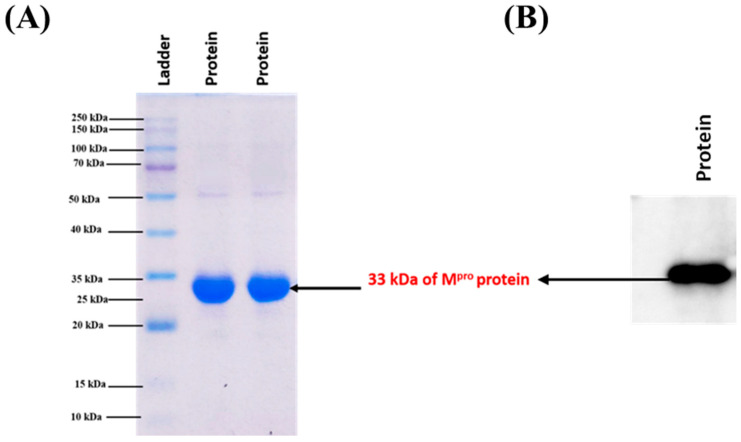
Electrophoretic analyses of purified M^pro^. (**A**) SDS-PAGE result showing a thick band of M^pro^ around 33 kDa. (**B**) A similar thick dark band can be seen around 33 kDa in the Western blot result.

**Figure 4 viruses-17-00402-f004:**
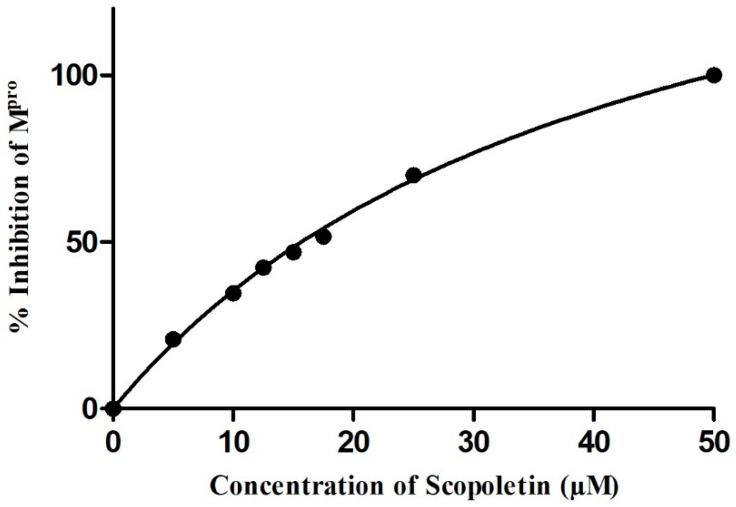
Enzymatic inhibition of M^pro^ in presence of different concentrations of Scopoletin; IC_50_ value was calculated as 15.75 µM.

**Figure 5 viruses-17-00402-f005:**
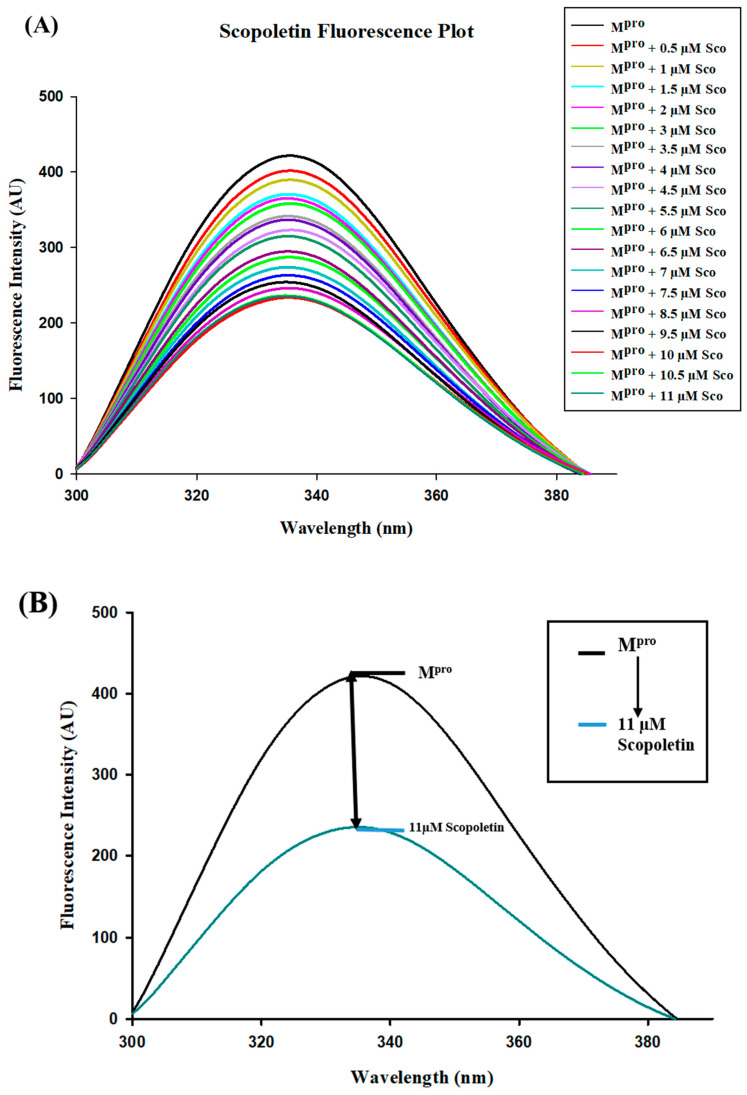
(**A**) Fluorescence emission spectra in the presence of varying concentrations of Scopoletin (0.5 µM to 11 µM). The progressive quenching of fluorescence intensity indicates the interaction between M^pro^ and Scopoletin. (**B**) Fluorescence emission spectra of M^pro^ in the native state and in the presence of an 11 µM concentration of Scopoletin. The significant reduction in fluorescence intensity highlights the quenching effect of Scopoletin on M^pro^. The arrow suggests the decrease in fluorescence intensity between the native and Scopoletin-bound states of M^pro^. Blue line indicates fluorescence quenching upon addition of 11 µM Scopoletin indicating physical interaction of this ligand with M^pro^.

**Figure 6 viruses-17-00402-f006:**
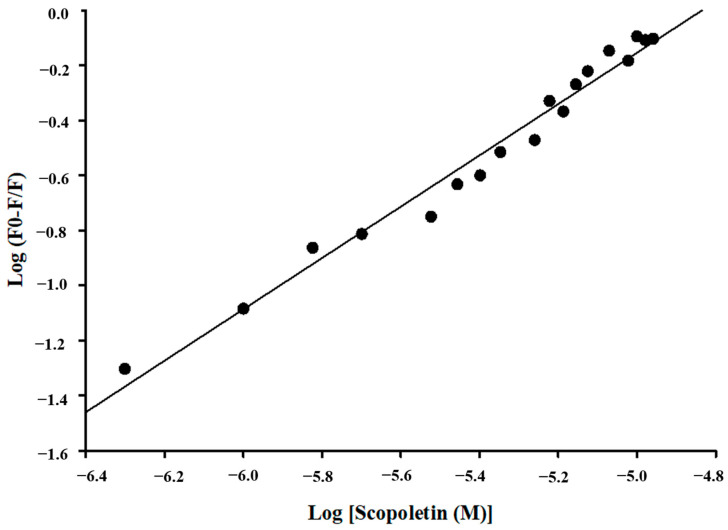
A graph representing the quenching of Scopoletin fluorescence is typically shown using a double logarithmic plot. The graph effectively illustrates the relationship between fluorescence quenching and the concentration of the quencher, Scopoletin.

**Figure 7 viruses-17-00402-f007:**
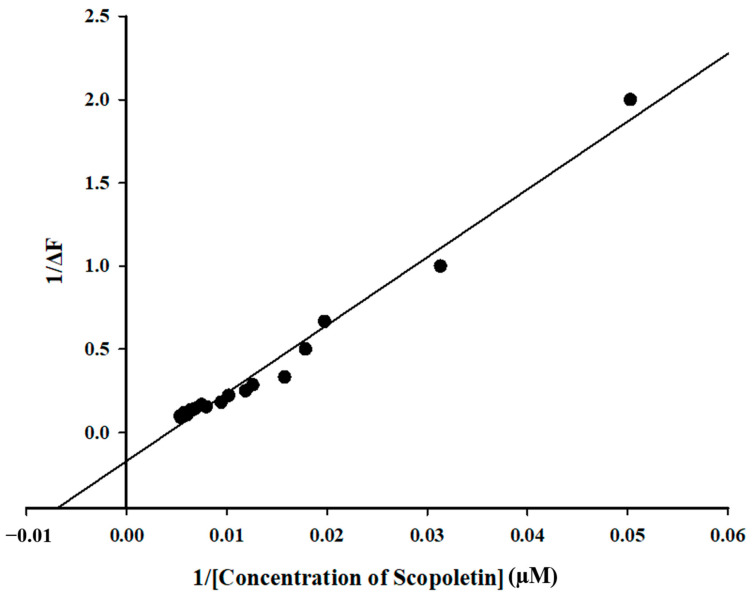
This plot involves graphing the reciprocal of the reaction rate (1/∆F) against the reciprocal of the ligand concentration (1/[concentration of Scopoletin] in µM) by examining the changes in the slope and intercepts of the plot in the presence of Scopoletin.

**Figure 8 viruses-17-00402-f008:**
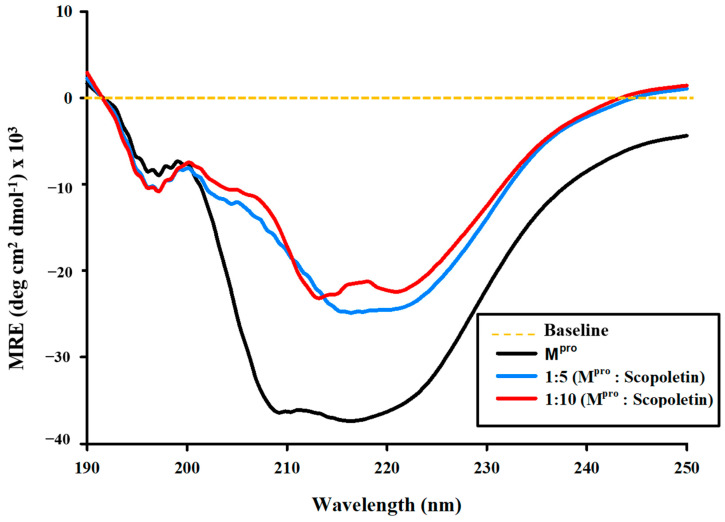
Graph depicting secondary structural changes in M^pro^ upon the addition of varying concentrations of Scopoletin. Although the overall shape remained the same, some reduction was observed in the secondary structure of M^pro^, which is indicative of specific and localized binding.

**Figure 9 viruses-17-00402-f009:**
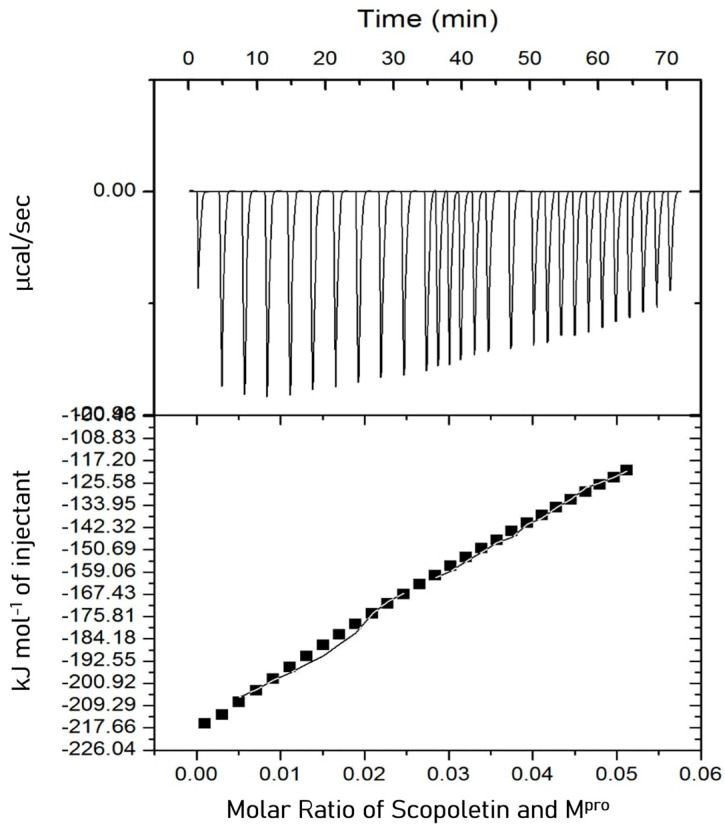
ITC profile of M^pro^ in the presence of Scopoletin. The raw data obtained by sequentially titrating Scopoletin into the sample cell containing the M^pro^ protein have been displayed in the upper panel. The binding isotherm displayed in the lower panels is generated by plotting the integrated heat results from the calorimetric titration after correcting for dilution heat versus the molar ratio of Scopoletin and M^pro^. Affinity/Kd = 4.24 × 10^−4^ M. 
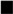
 boxes indicate change in heat energy in kJ mol^−1^ per injection of ligand to the protein.

**Figure 10 viruses-17-00402-f010:**
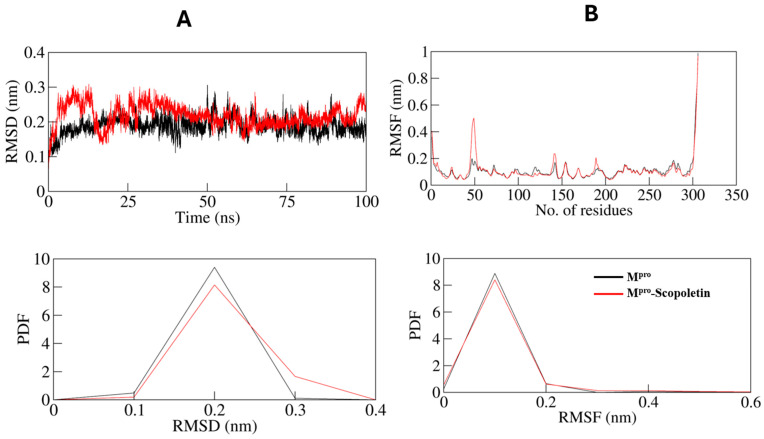
Compactness and structural dynamics of SARS-CoV-2 M^pro^ upon Scopoletin binding as a function of time. (**A**) RMSD profile of M^pro^ in complex with Scopoletin. (**B**) Residual fluctuations (RMSF) plot of M^pro^ before and after 0.107368 nm and 0.109368 nm. Black line: only M^pro^; red line: Scopoletin bound M^pro^.

**Figure 11 viruses-17-00402-f011:**
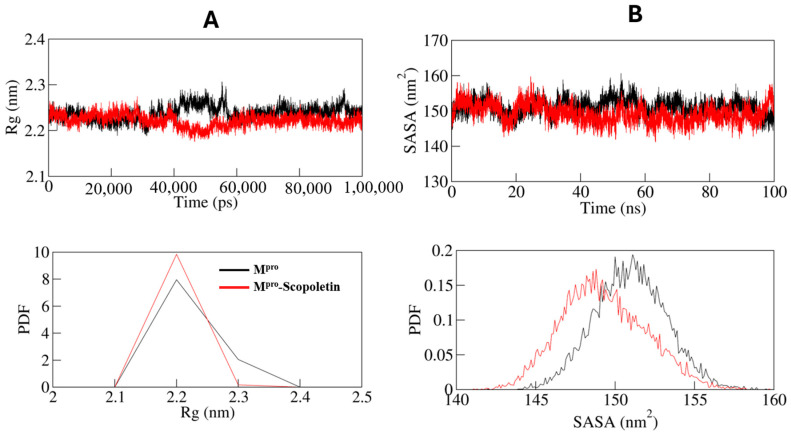
(**A**) Time evolution of radius of gyration. (**B**) SASA plot of M^pro^ as function of time. Black line: only M^pro^; red line: Scopoletin bound M^pro^.

**Figure 12 viruses-17-00402-f012:**
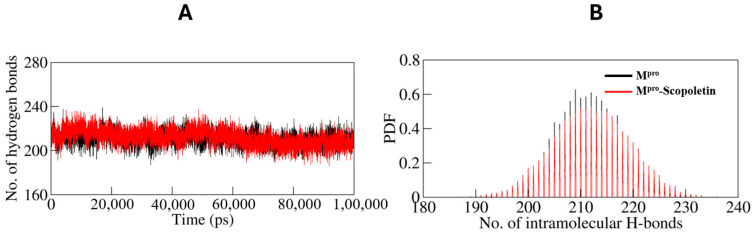
(**A**) Time evolution and stability of hydrogen bonds formed within 0.35 nm intra-M^pro^, and (**B**) the probability distribution function (PDF) of the H-bonds for both the systems. Black line: only M^pro^; red line: Scopoletin bound M^pro^.

**Figure 13 viruses-17-00402-f013:**
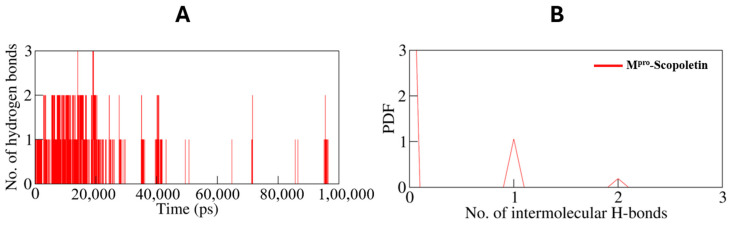
The simulation analyzed the stability of M^pro^ before and after Scopoletin binding by examining the time evolution of intramolecular hydrogen bonds (H-bonds) within 0.35 nm. The probability distribution function (PDF) of the H-bonds was also plotted, revealing average H-bond counts of 211 and 212 for the unbound and bound states, respectively.

**Figure 14 viruses-17-00402-f014:**
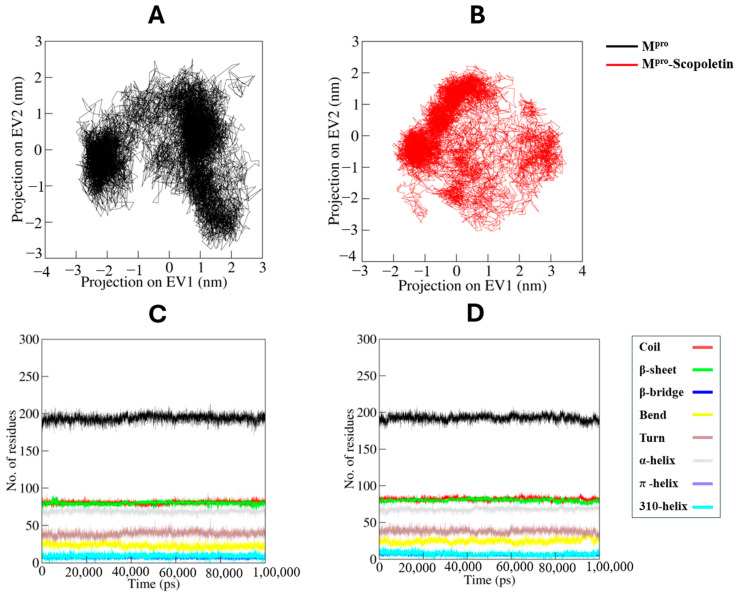
Principal component analysis. (**A**) Two-dimensional projections of trajectories on eigenvectors (EVs) showing conformational projections of SARS-CoV-2 M^pro^ and (**B**) M^pro^–Scopoletin (**C**) Time evolution of projections of trajectories on both EVs. (**D**) Residual fluctuations of M^pro^ on EV1.

**Figure 15 viruses-17-00402-f015:**
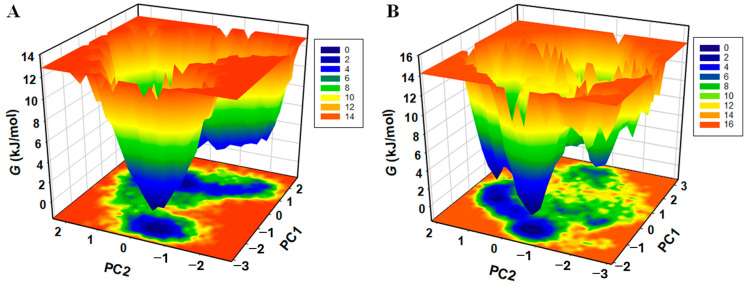
The Gibbs energy landscapes for (**A**) free M^pro^ and (**B**) M^pro^–Scopoletin.

**Figure 16 viruses-17-00402-f016:**
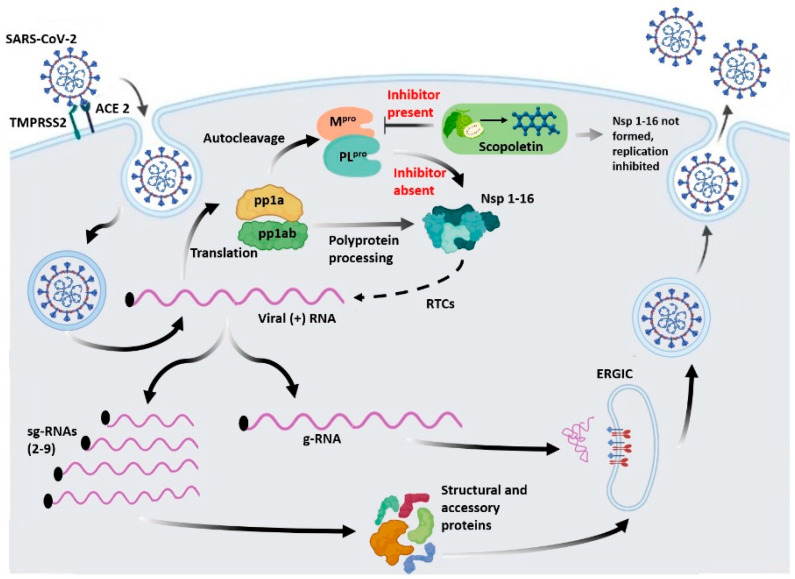
An illustration depicting the life cycle of SARS-CoV-2 in the presence and absence of the inhibitor Scopoletin. M^pro^, an essential cysteine protease in the life cycle of SARS-CoV-2, is responsible for polyprotein processing, which results in the formation of 16 non-structural proteins (nsps). These nsps are crucial for replicating and transcribing the viral genome. Scopoletin potentially inhibits M^pro^, thus impacting polyprotein processing and in turn inhibiting the replication and transcription of the viral genome.

**Table 1 viruses-17-00402-t001:** Binding energy scores of the five selected phytochemicals and positive control.

Ligand Name	Binding Free Energy (kcal/mol)	pKi	Ligand Efficiency (kcal/mol/non-H Atom)	Torsional Energy (kcal/mol)
**Scopoletin**	−5.5	4.03	0.3929	0.6226
**Gingerol**	−4.6	3.37	0.219	3.7356
**Pinine**	−4.9	3.59	0.49	0
**Limonene**	−4.9	3.59	0.49	0.3113
**Myrcene**	−3.9	2.86	0.39	1.2452
**Quercetin**	−7.1	3.89	0.3786	0.6226

**Table 2 viruses-17-00402-t002:** Physicochemical interactions of Scopoletin and Quercetin with SARS-CoV-2 M^pro^.

Compound and Its Chemical Structure	Conventional Hydrogen Bonds with M^pro^	Carbon Hydrogen Bonds with M^pro^	Van der Waals Force/Interaction/Bonds with M^pro^	P-alkyl/Alkyl Bonds with M^pro^
**Scopoletin** ** 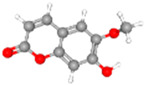 **	TYR54	HIS164	HIS41, GLY143, ARG188, ASP187, GLN189, GLU166, MET165	HIS163, CYS145, MET49
**Quercetin** ** 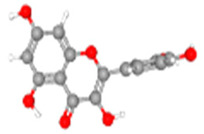 **	TYR54, THR26	Nil	ASN142, HIS41, PRO52, ARG188, ASP187, GLN189, MET165, GLU166, HIS164, LEU27, GLY143	MET49, CYS145

**Table 3 viruses-17-00402-t003:** ADMET properties of Scopoletin.

Compound ID	Absorption		Distribution	Metabolism	Excretion	Toxicity
	GI Absorption	Water Solubility	BBB permeation	CYP2D6Inh/Subs	OCT2 substrate	AMES
Scopoletin	High	Yes	Yes	No	No	No

**Table 4 viruses-17-00402-t004:** List of biological activities of Scopoletin, along with its plant sources, chemical name and structure.

Name of the Ligand	Plant Source	Ligand Structure	Chemical Name	Biological Activities	References
**Scopoletin**	Plants from Artemisia, Scopolia, Viburnum, and Mitracarpus genus.	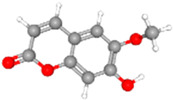	6-methoxy-7-hydrocoumarian	Antiviral Anti-cancer Antimicrobial Immunomodulatory Anti-angiogenesis Anti-oxidation Neuroprotective Anti-diabetic Antihypertensive Hepatoprotective Anti-inflammatory	[41], [70,71], PubChem CID: 5,280,460 [69]

## Data Availability

All associated data provided with the manuscript and its Appendix A.

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
