# Peer review of "Biochemical Screening of Phytochemicals and Identification of Scopoletin as a Potential Inhibitor of SARS-CoV-2 Mpro, Revealing Its Biophysical Impact on Structural Stability"

_viruses, 2025, doi:10.3390/v17030402_

Round 1
Reviewer 1 Report
Comments and Suggestions for Authors
In the manuscript (Bano et al, viruses-3448217), the authors have examined the potential role of Scopoletin as the inhibitor of the main protease (Mpro or 3CLpro or nsp5) of SARS-CoV-2. The study employed virtual screening and the resultant best compound, Scopoletin, was investigated by 1) fluorophore-based enzymatic assay, to quantify an experimental IC50 of 15.75 µM 2) fluorescence spectroscopy to show that Scopeoletin can quench trp fluorescence pf Mpro and 3) CD spectroscopy prominent changes in the helical content of Mpro after Scopoletin binding.
Altogether, this study is well designed and characterizes a previously identified (Sehercehajic et al) potential compound (Scopoletin) as an inhiubitor of SARS-CoV2 Mpro. However, the study lacks on several aspects and needs to address the following comments to substantiate their claim.
.
Comments:
- The authors do not compare the activity and efficiency of Querecitin in enzymatic assays before comparing its cytotoxic effect with Scopoletin. Likewise, other known Mro inhibitor could have been compared to understand the advantage of scopoletin over these inhibitors.
- The substrate used in the enzymatic assay is derived from a previous study (Rut W et al) which is still not peer reviewed. The reviewer could instead have used a natural specific substrate of SARS-Cov2 Mpro which is available commercially, The peptide MCA-AVLQSGFR-Lys(Dnp)-Lys-NH2 is a FRET substrate for the severe acute respiratory syndrome coronavirus main protease (SARS-CoV Mpro). The MCA-AVLQSGFR-Lys(Dnp)-Lys-NH2 substrate sequence is derived from residues P4–P5' of the SARS-CoV Mpro N-terminal autoprocessing site which has the sequence AVLQSGFRK.
- It would be of interest to know whether Scopoletin can show a similar effect on MPro of other members of the family.
- Some part of the text needs rewriting. For example. ( line 29-33, ;Utilizing.............presence of Scopoletin), there is no clarity about what the authors want to say.
NA
Reviewer 2 Report
Comments and Suggestions for Authors
The manuscript accounts of an interesting study demonstrating and analyzing the interaction of scopoletin with the main protease of SARS-CoV-2, resulting in inhibition of the activity of the protease and suggesting a possibility of a therapeutic application of this compound. The inhibition is rather weak (IC50 of ca 16 µM), which makes a direct therapeutic action improbable at scopoletin concentrations attainable in vivo but may suggest a search for scopoletin derivatives with higher inhibitory potency. Scopoletin has antioxidant properties, which could be of importance in ameliorating oxidative stress accompanying the SARS-CoV-2 infection. The binding energy, interactions and cytotoxicity of scopoletin is compared with quercetin; more comparisons could be made if the data for quercetin if available, especially the inhibition of protease activity.
The manuscript is somewhat untidy and would require an editorial and linguistic polishing.
Detailed remarks:
Graphical Abstract: The leave reminds more Cannabis than anything else, which may be misleading
Line 120-123: The statement is often but not generally true. Please consider, e.g., strychnine, currara or botulin
Lines 184/185: The values of R and G could be reported in SI units (J instead of cal)
Line 239: “-MCA” or “-AMC”?
Line 250: “at 380-455 nm wavelength”, not clear; are these excitation and emission wavelengths?
Wasn’t there any interference between the fluorescence of AMC and scopoletin?
Section 2.4.Please report sources and dilutions of antibodies. TBST, HRP, please explain the acronyms on the first use (though obvious for researchers employing blotting).
Lines 342/343: “The highest binding free energy was observed for Scopoletin”, after quercetin, which could be mentioned
Line 360 and Figure 2: “van der waal”, please correct to” van der Waals”
Section 3.3.What was the purity of the isolated protein?
Line 395: “at which 50% protein is inhibited”, rather: “at which 50% protein activity is inhibited”
Lines 453/454: Why different units are used (cal and J?)
Figure 9: Molar ratio of what to what?
Line 476: “no OCT2 substrate and AMES”, please be more informative,e.g.: “…is no OCT2 substate and shows no mutagenicity in the Ames test”
Line 468: “does not have CYP2D6Inh/Subs”, please state more clearly
Page 26: Please provide the caption to the Table
The paper of Yuan et al. (https://doi.org/10.1016/j.lfs.2021.119105) supports some conclusions of the authors (to make it clear: I have no relation with this paper)
